# Effectiveness of COVID-19 Vaccines against Delta Variant (B.1.617.2): A Meta-Analysis

**DOI:** 10.3390/vaccines10020277

**Published:** 2022-02-11

**Authors:** Rashidul Alam Mahumud, Mohammad Afshar Ali, Satyajit Kundu, Md Ashfikur Rahman, Joseph Kihika Kamara, Andre M. N. Renzaho

**Affiliations:** 1NHMRC Clinical Trials Centre, Faculty of Medicine and Health, The University of Sydney, Camperdown, NSW 2006, Australia; 2School of Business and Centre for Health Research, University of Southern Queensland, Toowoomba, QLD 4350, Australia; mohammadafshar.ali@usq.edu.au; 3Health Research Group, Department of Statistics, University of Rajshahi, Rajshahi 6205, Bangladesh; 4Allied Health and Human Performance Unit, University of South Australia, Adelaide, SA 5001, Australia; 5Department of Economics, Jagannath University, Dhaka 1100, Bangladesh; 6Faculty of Nutrition and Food Science, Patuakhali Science and Technology University, Patuakhali 8602, Bangladesh; satyajit@seu.edu.cn; 7School of Public Health, Southeast University, Nanjing 210009, China; 8Development Studies Discipline, Khulna University, Khulna 9208, Bangladesh; ashfikur@ku.ac.bd; 9World Vision East Africa Regional Office, Karen, Off Ngong Road, Karen Road, Nairobi 50816, Kenya; joseph_kamara@wvi.org; 10Centre for Humanitarian Leadership, Deakin University, Melbourne, Burwood, VIC 3125, Australia; 11School of Medicine, Translational Health Research Institute, Western Sydney University, Campbelltown, NSW 2150, Australia; andre.renzaho@westernsydney.edu.au; 12Maternal, Child and Adolescent Health Program, Burnet Institute, Melbourne, VIC 3004, Australia

**Keywords:** delta variant, COVID-19 vaccines, herd immunity, vaccines effectiveness, meta-analysis, B.1.617.2

## Abstract

**Objectives:** The highly transmissible COVID-19 Delta variant (DV) has contributed to a surge in cases and exacerbated the worldwide public health crisis. Several COVID-19 vaccines play a significant role in a high degree of protection against the DV. The primary purpose of this meta-analysis is to estimate the pooled effectiveness of the COVID-19 vaccines against the DV in terms of risk ratio (RR) among fully vaccinated, compared to unvaccinated populations. **Methods:** We carried out a systematic review, with meta-analysis of original studies focused on COVID-19 vaccines effectiveness against a DV clinical perspective among fully COVID-19 vaccinated populations, compared to placebo (unvaccinated populations), published between 1 May 2021 and 30 September 2021. Eleven studies containing the data of 17.2 million participants were identified and included in our study. Pooled estimates of COVID-19 vaccines effectiveness (i.e., risk ratio, RR) against the DV with 95% confidence intervals were assessed using random-effect models. Publication bias was assessed using Egger’s regression test and funnel plot to investigate potential sources of heterogeneity and identify any differences in study design. **Results:** A total population of 17.2 million (17,200,341 people) were screened for the COVID-19 vaccines’ effectiveness against the DV. We found that 61.13% of the study population were fully vaccinated with two doses of COVID-19 vaccines. The weighted pooled incidence of COVID-19 infection was more than double (20.07%) among the unvaccinated population, compared to the fully vaccinated population (8.16%). Overall, the effectiveness of the COVID-19 vaccine against the DV was 85% (RR = 0.15, 95% CI: 0.07–0.31). The effectiveness of COVID-19 vaccines varied slidably by study designs, 87% (RR = 0.13, 95% CI: 0.06–0.30) and 84% (RR = 0.16, 95% CI: 0.02, 1.64) for cohort and case-control studies, respectively. **Conclusions:** The effectiveness of COVID-19 vaccines were noted to offer higher protection against the DV among populations who received two vaccine doses compared with the unvaccinated population. This finding would help efforts to maximise vaccine coverage (i.e., at least 60% to 70% of the population), with two doses among vulnerable populations, in order to have herd immunity to break the chain of transmission and gain greater overall population protection more rapidly.

## 1. Introduction

On 19 May 2021, the Delta variant (DV) of concern, formerly known as the Indian variant of concern (or B.1.617.2), became the most dominant strain of severe acute respiratory syndrome coronavirus 2 (SARS-CoV-2) [1]. The SARS-CoV-2 DV that was reported in 185 countries [2] is more contagious, with a higher number of R-naught (R_0_ of 5–6) compared to other variants (e.g., Alpha, Beta, Gamma) [3]. A significant number of new COVID-19 cases, caused by the highly transmissible DV, is exacerbating the worldwide public health crisis [4]; 99% of sequenced and genotyped cases are related to the DV. The burden of the DV has led to considerable demand for herd immunity and the optimal timing of booster doses for the vaccinated population [5]. Coronavirus disease-19 (COVID-19) vaccines have been identified as the most effective means of halting the spread of the disease over the world. Several vaccines are currently being administered to people of various ages all around the world. Nonetheless, two doses of vaccines have played a significant role in terms of a high degree of protection before the Delta variant than one dose of vaccination [6]. The vaccines’ performance remains significant, but they have yet to lead to herd immunity to break the chain of transmission of the Delta variant [7]. Emerging evidence suggests the BNT162b2 mRNA vaccine (Pfizer-BioNTech) and the Oxford AstraZeneca adenovirus-vector vaccine, ChAdOx1 nCoV-19 (ChAdOx1), exhibit great efficiency against the Alpha (B.1.1.7) variant and other variants before the DV [8,9,10,11].

The continued appearance of novel SARS-CoV-2 variations could jeopardise the efficacy of immunisation efforts, especially since in vitro tests suggest that vaccine-elicited antibodies have lower neutralisation activity against emergent variants [12,13]. The DV is of particular concern as it has resulted in substantial increases in infections in many countries, including ones with excellent vaccination coverage, such as the United Kingdom (UK), where it swiftly became dominant after being recognised as a Variant of Concern (VOC) on 28 April 2021 [13].

The rapid pace at which data is emerging, as well as heterogeneities in study protocols and designs, different socioeconomic and demographic characteristics (i.e., age, sex, ethnicity, comorbidities), and regions and implementation, make interpreting and assessing the anticipated impact of vaccine campaigns on local epidemics difficult and challenging. Measuring the effectiveness of COVID-19 vaccines against the most dominant variant (B.1.617.2) can be a basis for accentuating protection in patients with COVID-19 infection. In contrast, there is still a scarcity of real-world data on the effectiveness of vaccines against the DV [14]. Therefore, using a systematic review and meta-analysis, we aim to estimate the pooled COVID-19 vaccines efficacy and effectiveness against the DV. This study is important and, unlike previous systematic reviews and meta-analyses [15,16,17], which have focused on risk assessment, immunogenicity, reactogenicity and efficacy across different vaccine platforms.

We anticipated that our study findings will provide a comprehensive understanding of the effectiveness of COVID-19 vaccines and will inform strategies to promote protection against COVID-19, especially the DV. Furthermore, the study provides real-time evidence to boost vaccine confidence. Globally, there have been unfounded concerns that have affected public confidence in vaccines and triggered vaccine hesitancy, which could delay the fight against the pandemic if not urgently addressed [18,19].

## 2. Methods

### 2.1. Search Strategy

We conducted a comprehensive search for academic research studies that reported the effectiveness of COVID-19 vaccines against the COVID-19 Delta variant, published between 1 May 2021 and 30 September 2021, from the following electronic bibliographic databases: Scopus, EBSCOhost (CINAHL, Medline), PubMed, Web of Science, and the first 20 pages of Google Scholar. The following search terms were used with no language restrictions:

[(“Effectiveness” OR “Efficacy” OR “Security” OR “Safety” OR “Evaluation”) AND (“COVID-19 Vaccines” OR “Coronavirus-19 Vaccines” OR “mDNA COVID-19 vaccines” OR “Moderna” OR “mRAN-1273” OR “Pfizer/BioNTech” OR “BNT162b2” OR “Johnson & Johnson” OR “Ad26.COV2.S” OR “Oxford/AstraZeneca” OR “AZD1222” OR “BBIBP-CorV” OR “Vero Cells” OR “CoronaVac” OR “Vaccines”) AND (“Delta Variant” OR “B.1.617.2” OR ““Delta” OR “Alpha Variant” OR “Variant”)].

#### 2.1.1. Study Participants

This meta-analysis study comprises vaccinated or unvaccinated populations that were infected with COVID-19 since the delta variant emerged.

#### 2.1.2. Inclusion and Exclusion Criteria

In this meta-analysis, eligible studies were selected if they (1) were original articles; (2) published between 1 May 2021 and 30 September 2021 (focused on the Delta variant); (3) focused on the effectiveness of COVID-19 vaccines against the DV (B.1.617.2) clinical perspective, (4) effectiveness or efficacy of all COVID-19 vaccines, (5) effectiveness or efficacy on fully COVID-19 (i.e., two-doses) vaccinated people compared to placebo (unvaccinated), and (6) reported clinical characteristics of COVID-19 among infected people. Editorials, letters, perspectives, commentaries, reports, opinion species, reviews, and studies with ‘insufficient related data’ were excluded from the analysis. The reference lists of studies included were checked for eligible studies. We selected the starting date of 1 May 2021 because it marks a period when B.1.617.2 was escalated, the VOC having been earlier detected in April 2021 [13,20].

#### 2.1.3. Data Extraction

Data were independently extracted into EndNote libraries by three researchers who later compared their results. Emerging differences in the data were discussed and resolved by consensus, and, where the three researchers could not agree, the fourth and fifth researchers were consulted for adjudication. The extracted data captured the author’s name, year, settings, design or approach, the overall number of COVID-19 infections, the number of vaccinated and unvaccinated participants, and the number of participants.

#### 2.1.4. Study Screening and Selection

The screening and selection of studies followed a three-stage screening procedure. The first stage involved screening studies by title to eliminate duplicates. The second stage required the reading of abstracts to determine their relevance to our study. The third stage necessitated the reading of full texts of the retained studies from the second stage, and those that met the set criteria were retained for our study, as reflected in Figure 1. R.A.M. carried out and recorded the above process and shared the record with two other authors (M.A.A. and M.A.R.) for verification. Discrepancies were discussed and resolved by consensus.

#### 2.1.5. Quality Assessment

Quality assessment was independently conducted by two authors who applied four quality assessment tools due to the heterogeneity of included studies designs. Discrepancies in the two authors’ quality assessments were referred to the other authors for adjudication. The included studies designs were cohort, case series, and case controls. The three critical appraisal tools applied were from the Joanna Briggs Institute (JBI). The JBI tools have been used widely in academic studies [21,22]. The tools provide a subjective assessment of risk of bias (ranked as low, moderate, or high) [23]. Higher quality indicates greater confidence that future research is unlikely to change or contradict the results, while lower quality indicates a higher likelihood that future research may not affirm the study results. The first JBI tool used was the checklist for cohort studies, which we applied to four studies [24,25,26,27] included in this systematic review and meta-analysis. The checklist assesses critical areas of studies’ methodologies for biasness in design, implementation, and analysis (Appendix A
Table A1). The second tool of quality assessment used was the JBI checklist for analytical case–control studies, which we applied to the six case–control studies [6,28,29,30,31,32] included in this review. The checklist consists of 10 items that enable the critical appraisal of studies for potential biasness (Appendix A
Table A2). The third and last tool used was the JBI checklist for case series studies, which was applied to one study included [33] in our review (Appendix A
Table A3). The checklist comprises of 10 items measuring the comparability of the study groups, the matching of cases and controls, the identification criteria for cases and controls, the reliability and validity of exposure measurement, similarities in exposure measurement for case and control, the identification of confounding factors, addressing the confounding factors, the assessment of outcomes, the length of the exposure period, and statistical analysis.

### 2.2. Data Analysis

Given the high heterogeneity between studies (I^2^ > 50%) [34], we used random effects meta-analysis models to obtain pooled two-dose COVID-19 vaccines effectiveness (VE) estimates against the Delta variant (B.1.617.2), complemented with a sensitivity analysis to examine the effects of outliers. VE was calculated as 1—risk ratio (RR).
$$\mathrm{VE}=\frac{\left(\mathrm{Risk}\;\mathrm{among}\;\mathrm{unvaccinated}\;\mathrm{group}-\mathrm{Risk}\;\mathrm{among}\;\mathrm{vaccinated}\;\mathrm{group}\right)}{\mathrm{Risk}\;\mathrm{among}\;\mathrm{unvaccinated}\;\mathrm{group}}$$
Alternatively, VE=1−risk ratio (RR)

For two-dose VE, we calculated the pooled estimate of VE against the delta variant and by study design. We assessed residual heterogeneity between studies by calculating the I^2^ statistic [34]. The I^2^ statistic enabled us to determine whether the percentage of variance was attributable to the heterogeneity of the data in studies included using the random-effects model. We used forest plots to show the distribution of RR of COVID-19 infection during the delta variant. Subgroup analysis based on the vaccine phases significantly decreased the heterogeneity in the high heterogeneity cases. The presence and effect of publication bias were assessed by using Egger’s regression test [35] and funnel plot [36] to investigate potential sources of heterogeneity and identify any differences in study design. All analyses, two-tailed statistics and a significance level of less than 0.05 were considered. All statistical analyses were performed by STATA/SE version 15.0 (StataCorp, College Station, TX, USA).

## 3. Results

### 3.1. Description of Studies Included

Our primary search of databases yielded 753 studies, of which 11 met the criteria (i.e., six studies were case-control [6,28,29,30,31,32], four studies were cohort design [24,25,26,27], and one study was case-case design [33]) and were included in this study (Figure 1). The included studies had a total sample of ~17.2 million (N = 17,200,341) people and, of these, 61.13% of were fully vaccinated (*n* = 10,514,132) with two-doses of COVID-19 vaccines (Table 1). The COVID-19 infection was more than double (21.61%, 95% CI: 8.50%–34.72%) among the unvaccinated population, compared to the fully vaccinated population (8.88%, 95% CI: 0.18%–17.95%).

### 3.2. Effectiveness of the COVID-19 Vaccines against the Delta Variant

Figure 2 shows the effectiveness of the COVID-19 vaccines against the Delta variant after two doses. All four cohort studies showed that vaccines played a significant role in reducing the risk of COVID-19 infections among the fully vaccinated population, compared to the unvaccinated population. The pooling of four cohort studies [24,25,26,27] showed that vaccines significantly lowered the risk (RR = 0.20, 95% CI: 0.07–0.54) of exposure against the DV among the fully vaccinated (i.e., two-doses) population by 80% compared to the unvaccinated population. Pooled estimates for six case–control studies, except one study, showed a significantly lower risk of having COVID-19 infection (90%) [RR = 0.10, 95% CI: 0.01–0.80] against the delta variant in the fully vaccinated population than the unvaccinated population (Figure 2). Overall, the effectiveness of COVID-19 vaccines against the delta variant was 86% (RR = 0.14, 95% CI: 0.07–0.54). The funnel plot shows some asymmetry (Figure 3); however, this is not significant for Begg’s regression test (*p* = 481 for five case–control studies and *p* = 0.314 for five cohort studies) (Table 2), denoting an absence of publication bias. Therefore, we included all 11 studies in the meta-analysis. Our analysis suggests statistically significant heterogeneity (I^2^ = 99.9%, *p* < 0.001 for six case–control studies and I^2^ = 99.8%, *p* < 0.001 for four cohort studies, I^2^ = 99.8%, *p* < 0.001) (Figure 2).

#### Quality of Included Studies

Our findings suggest that five (*n* = 5) of the included studies [6,28,29,30,33] were of high quality, implying that they were robust studies; six of the studies [24,25,27,29,31,32] were ranked as medium quality, implying that their methods were of moderate quality based on the JBI quality assessment criteria. There was no study excluded based on poor scoring on the quality assessment scales. Notably, one (*n* = 1) cohort study [26] was assessed as high on the JBI tools; the rest (*n* = 3) cohort studies were found to be of medium quality on the same scale [24,25,27]. Three out of the six (*n* = 3/6) case–control studies included [6,28,30] were found to be of high quality based on the JBI quality assessment scale for case–control studies; the rest (*n* = 3) of the studies [29,31,32] were found to be of medium quality on the same scale (see Appendix A Table A1, Table A2 and Table A3 for details). Lastly, the only case-series study [33] included in our review was found to be of high quality on the JBI quality assessment scale of case controls.

## 4. Discussion

In this catastrophic situation, several vaccines, including Pfizer-BioNTech, Moderna, and AstraZeneca, protect against SARS-CoV-2. These vaccines teach our cells how to make a protein (or even just a piece of a protein) that triggers an immune response inside our bodies. Theoretical benefits of these vaccines are that the vaccinated population will gain protection without ever having to risk the serious consequences of getting sick and dying of COVID-19. Several studies have confirmed that these vaccines trigger an immune response against the virus [26,37,38]. The vaccines have been approved for clinical use by the World Health Organization based on comprehensive assessments of non-clinical, laboratory confirmation, clinical utility, and manufacturing data provided by manufacturers to respective regulatory bodies (i.e., US Food and Drug Administration (FDA) and the Department of Health and Human Services or related institutes). Though these vaccines’ efficacy and side effects have not yet been widely discussed, numerous claims have been made by popular media and politicians [39,40]. During this process, respective government bodies examined the quality and consistency of the vaccine to determine if the potential benefits offset the potential adverse effects of the vaccine. Despite unsubstantiated claims on the COVID-19 vaccines’ efficacy and associated side effects, there is a lack of accumulated evidence on the effectiveness of this vaccine [26]. However, among different variants of SARS-CoV-2, the DV has a significantly increased virulence and transmission capacity [41]. In addition, recent evidence indicates that the vaccine’s effectiveness declined with the emergence and growth of the DV [6]. Therefore, this meta-analysis presents a piece of systematic and evidence-based information regarding the effectiveness of different COVID-19 vaccines against the specific DV.

The DV has seen a sharp rise in infections and mortality in 185 countries [2], including those with relatively high vaccine coverage and increased hospitalisation and mortality in the UK [20,42]. The high transmissibility of SARS-CoV-2, mainly of the DV, is a considerable socioeconomic global burden. Previous evidence on vaccine effectiveness reported reduced protection against the DV compared to the Alpha variant [1,30]. Even the efficacy of the two-dose ChAdOX1 vaccine decreased by 59.8% after exposure to B.1.617.2 [30]. Recent reports determined a higher vaccine breakthrough infection against the SARS-CoV-2 DV [1,30]. While more contributing factors may reduce immunity; a recent Israeli study has found higher infection rates even among the earliest vaccinated individuals [43]. Another study among nursing home residents found the effectiveness of the mRNA vaccine as 74.7% against infection, but the efficiency decreased significantly to 53.1% when the B.1.617.2 circulation predominated [27]. The interpretation of this finding is complex due to the bias of the methodology, research population, care settings, immunisation timeline and the potential selection and case diagnosis [6]. Our study found that the effectiveness of the COVID-19 vaccines was 80% more protective against the DV (RR = 0.20, 95% CI: 0.07–0.54) among those vaccinated compared to the unvaccinated population. A previous study also found a higher rate of infection, indicating the lower effectiveness of vaccines in protecting infection with the DV [33]. Another study also argued that infections with the DV that are transmitted despite the vaccine carry a similar viral burden for unvaccinated people [14]. Though the effectiveness of vaccines is lower among DV cases, completing the second dose of vaccines reduces the complication among those vaccinated [6]. In addition, a growing number of cases detected after one or two doses of vaccination are more likely to be infected with the DV than other variants [12]. Since the efficacy of the vaccines against the DV is lower, several studies have recommended an additional dose to curb the infection of the variant and optimise the immune response [12,27]. Single-dose vaccine recipients with high infection rates of DV or who have been infected before with another variant can risk vaccine-adapted forms of evolution [12]. We argue that the population should be vaccinated as much as possible since the unvaccinated cannot be protected by a significant reduction in the number of vaccinated people, as seen in other infections, making herd immunity unachievable due to emerging variants [14].

Our study is unique, with several strengths. Firstly, it is the first meta-analysis that provides evidence-based data on vaccines’ effectiveness against the DV. Secondly, we only considered studies that provide symmetrical information of individuals vaccinated with two doses. Thirdly, a large population was considered in this study, yielding adequate evidence on this topic. However, we acknowledge some limitations with our study design and selection of published studies. Effectiveness of vaccines (e.g., BNT162b2 or ChAdOx1 nCoV-19) after one dose was significantly lower among the population with the DV (30.7%) than among those with the Alpha variant (48.7%) [30]. Studies on single-dose vaccination and its efficacy were excluded in this meta-analysis due to the lack of reported data, lower effectiveness with the DV, and an insufficient number of studies, and we did not employ any analysis considering the sub-groups of different types of vaccines due to the insufficient number of studies that reported combined results with more than one vaccine.

Nonetheless, our study provides valuable insights into the effectiveness of the vaccines against the DV since there is a dearth of information on this topic. Depending on the predominance of the DV, the findings of this study may help policy-makers to act or undertake control measures and coordinate vaccination efforts. Recently, caution has been elevated that vaccine performance may diminish in the future, considering the possibility of a series of potential viral genetic drifts [6]. These claims highlight the importance of both pharmaceutical and non-pharmaceutical interventions in tandem in order to reduce SARS-CoV-2 infection. Future research could examine the combined efficacy of mass vaccination, contact tracing and social distancing measures in minimising the socioeconomic burden arising from COVID-19. Our study’s practical implication is that taking the mounting evidence on the efficacy of pharmaceutical intervention into account, mass vaccinations should be continued in a planned manner to lessen the socioeconomic burden of COVID-19. To do so, COVID-19 vaccines should be categorised as a global humanitarian good, emphasising the optimisation of production and equal distribution. On the other hand, the demand side should target reducing vaccine hesitancy so that more vulnerable populations are adequately covered.

## 5. Conclusions

The effectiveness of COVID-19 vaccines was noted to offer higher protection against the Delta variant among populations after receiving two doses compared with unvaccinated populations. This finding would help efforts to maximise vaccine coverage (i.e., at least 60% to 70% of the population), with two doses among vulnerable populations, in order to create herd immunity, break the chain of transmission and gain greater overall population protection more rapidly.

## Figures and Tables

**Figure 1 vaccines-10-00277-f001:**
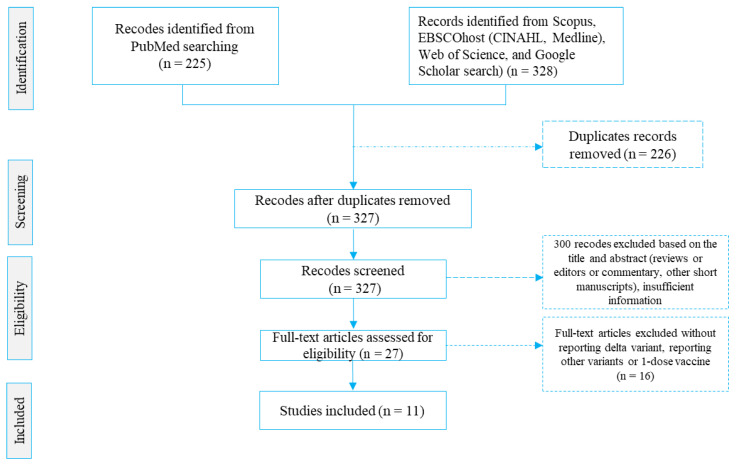
Steps of study selection procedures.

**Figure 2 vaccines-10-00277-f002:**
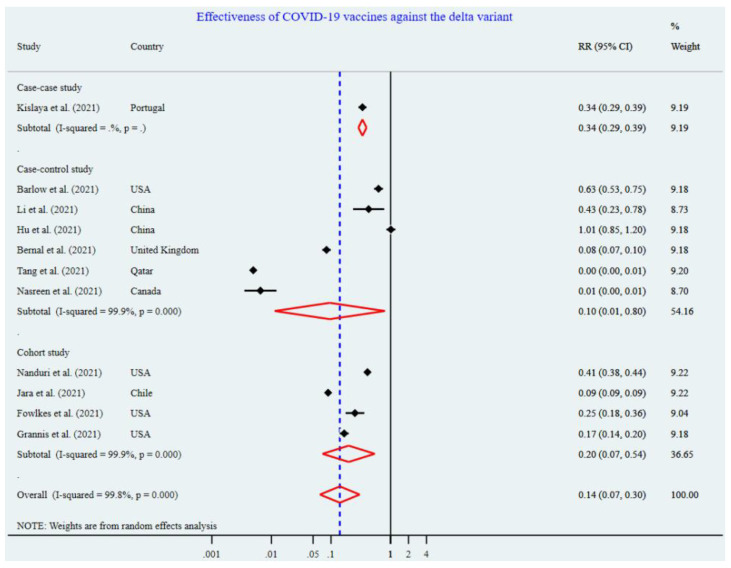
Effectiveness of the two-dose COVID-19 vaccines against the Delta variant.

**Figure 3 vaccines-10-00277-f003:**
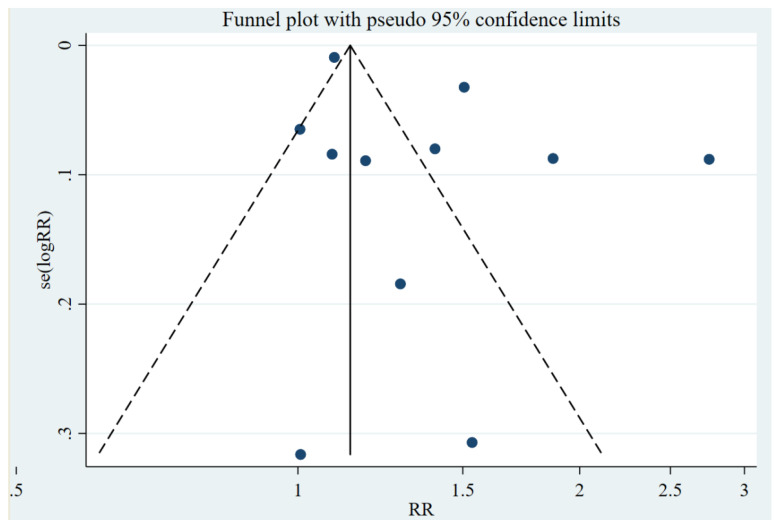
Assessing publication bias.

**Table 1 vaccines-10-00277-t001:** Characteristics of the selected studies.

Study	Country	Design	Fully Vaccinated Group		Unvaccinated Group	
			Number of Infected Persons, *n* (%)	Total Vaccinated Population	Number of Infected Persons, *n* (%)	Total Unvaccinated Population
Bernal et al. (2021) [30]	UK	Case–control	143 (0.60)	23,993	7313 (7.59)	96,371
Fowlkes et al. (2021) [24]	USA	Cohort study	34 (1.14)	2976	194 (4.69)	4136
Grannis et al. (2021) [25]	USA	Cohort study	134 (2.94)	4551	1185 (20.76)	5708
Barlow et al. (2021) [6]	USA	Case-control	145 (29.00)	500	279 (55.80)	500
Hu et al. (2021) [29]	China	Case–control	187 (39.29)	476	184 (38.66)	476
Jara et al. (2021) [26]	Chile	Cohort study	12,286 (0.29)	4,173,574	185,633 (3.39)	5,471,728
Kislaya et al. (2021) [33]	Portugal	Case-case study	162 (11.89)	1363	777 (46.00)	1689
Li et al. (2021) [28]	China	Case–control	12 (12.37)	97	37 (34.58)	107
Nanduri et al. (2021) [27]	USA	Cohort study	2999 (0.06)	5,011,746	1397 (0.15)	953,861
Nasreen et al. (2021) [32]	Canada	Case–control	10 (0.12)	8461	6325 (22.03)	28,705
Tang et al. (2021) [31]	Qatar	Case–control	249 (0.02)	1,286,395	4993 (4.06)	122,928
Total, *n* = 12			N = 16,361 (0.16)	N = 10,514,132	N = 208,317 (3.12)	N = 6,686,209
Weighted pooled incidence of COVID-19 infection.			8.88% (95% CI: 0.18–17.95)		21.61% (95% CI: 8.50–34.72)	

**Table 2 vaccines-10-00277-t002:** Assessing publication bias.

Study Design	Number of Studies	Egger’s Regression Test				Small-Study Effect (*p*-Value)
		RR, (95% CI)	*p*-Value	Bias (RR), 95% CI	*p*-Value	
Case-control	6	0.61 (0.02, 5.53)	0.224	30.53 (0.89, 51.45)	0.481	*p*-value = 0.685
Cohort study	4	0.09 (0.04, 0.17)	0.015	13.32 (0.98, 45.47)	0.390	*p*-value = 0.390
Case-case study	1	-	-	-	-	-

RR = risk ratio, *p*-value = probability value.

## Data Availability

The present study was a meta-analysis based on published original research articles. All data are publicly available within the manuscript. We have accessed the published articles. All statistical analyses were performed by STATA/SE version 15. The corresponding author can access a custom analysis codes (stata do-file).

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
