# Peer review of "Effectiveness of COVID-19 Vaccines against Delta Variant (B.1.617.2): A Meta-Analysis"

_vaccines, 2022, doi:10.3390/vaccines10020277_

Round 1
Reviewer 1 Report
I think it is an interesting work on COVID-19 vaccines and will bring clarity and arguments to those who read and use it.
The authors establish in the title the type of design they use, but later, in the "Methods" section, they do not mention it explicitly, something that they do in the abstract. From my point of view, it should be mentioned in "Methods" as well.
In the "Methods" section, subsection 2.1.1. "study participants" (page 3) , I think the sentence could be improved if instead of saying "during the delta variant" it was said "since the delta variant emerged".
In subsection 2.1.4. "study screening and selection" (page 3), two acronyms are used (RAM and MAR) which, as I have been able to deduce, correspond to the initials of two of the authors of the manuscript. To avoid confusion, it should be made more explicit.
Author Response
Response to the Reviewer-1 comments
- I think it is an interesting work on COVID-19 vaccines and will bring clarity and arguments to those who read and use it.
Authors’ response: We express our gratitude to the reviewer’s affirmation of the study’s novelty and strength.
- The authors establish in the title the type of design they use, but later, in the "Methods" section, they do not mention it explicitly, something that they do in the abstract. From my point of view, it should be mentioned in "Methods" as well.
Authors’ Response: Thank you. We have revised the methods section to add the study design. Please see the revised methods section on page 3.
- In the "Methods" section, subsection 2.1.1. "study participants" (page 3) , I think the sentence could be improved if instead of saying "during the delta variant" it was said "since the delta variant emerged"
Authors’ response: Thank you. The sentence has been revised. Now it reads: “This meta-analysis scope comprises of vaccinated or unvaccinated populations that were infected with COVID-19 since the delta variant emerged.” Please see page 3 (2.1.1 sub-section).
- In subsection 2.1.4. "study screening and selection" (page 3), two acronyms are used (RAM and MAR) which, as I have been able to deduce, correspond to the initials of two of the authors of the manuscript. To avoid confusion, it should be made more explicit.
Authors’ Response: Thank you for picking these up. We have revised this sentence. Now it reads: “RAM carried out and recorded the above process and shared the record with other two authors (MAA and MAR) for verification. Discrepancies were discussed and resolved by consensus”. Please see page 4.
Reviewer 2 Report
This is very interesting manuscript with meta analysis on the effectiveness of COVID-19 delta variant on non vaccinated and vaccinated population. The number size is good enough to conclude the results. I have some minor comments-
- Is it possible to apply computational or statistical measurements to show the effectiveness of delta variant on both vaccinated and non vaccinated population through global mapping?
- Also, an inclusion of a table based on which vaccine has been reported more effective against delta variant? Because this one of the great discussion now days about the vaccine effectiveness.
- It would be nice to add a short study on the effects of delta variant in post vaccinated population (prefer to present data through mapping).
Overall, manuscript is well written and data also presented quite well. Possibly after these changes manuscript would be highly interesting for the readers.
Author Response
Cover Letter
Date: February 4, 2022
Dear Ms. Cynthia Wang
Editor,
Vaccines
Manuscript Number: vaccines-1583138
The revised title of the manuscript: "Effectiveness of COVID-19 vaccines against delta variant (B.1.617.2): A meta-analysis".
Dear Cynthia,
Thank you for the opportunity to revise our manuscript. We found the reviewers' comments and feedback very helpful in improving the manuscript and we have revised the manuscript accordingly. Please find to follow a point by point responses to each of the reviewers' comments. Two versions of our paper - one clean copy, and one marked copy showing the changes made are submitted.
We look forward to hearing from you.
Yours sincerely,
Dr Rashidul Alam Mahumud (corresponding author)
On behalf of all co-authors
NHMRC Clinical Trials Centre,
The University of Sydney,
Australia
Email: rashed.mahumud@sydney.edu.au
Phone: +61 2 9562 5294
Fax: +61 2 9565 1863
Response to the editor comments:
- Please note that author names, affiliations and e-mail could not be changed if paper accepted, so please check it carefully when revising your manuscript.
Authors’ Response: We have checked all authors’ names and affiliations and can confirm they are correct.
- Your manuscript has been reviewed by experts in the field. Please find your manuscript with the referee reports at this link:
https://susy.mdpi.com/user/manuscripts/resubmit/2851d15faacd3356c913a7eefc13e6db
(I) Please revise your manuscript according to the referees’ comments and upload the revised file by 2 February.
(II) Please use the version of your manuscript found at the above link for your revisions.
(III) Any revisions made to the manuscript should be marked up using the “Track Changes” function if you are using MS Word/LaTeX, such that changes can be easily viewed by the editors and reviewers.
(IV) Please provide a short cover letter detailing your changes for the editors’ and referees’ approval.
Authors’ response: Thank you for the opportunity to revise our manuscript. We found the reviewers' comments and feedback very helpful in improving the manuscript and we have revised the manuscript accordingly.
- During our initial check, we noticed that section "conclusion" is missing.
As this section is necessary for publication, please add it in your manuscript.
Authors’ Response: Thank you. We have added the conclusion section. Please see the revised manuscript.
- Please keep one first author.
The journal is now investigating multiple first co-authors issues, you are suggested to keep such as only one first author based on the real arrangement of your research project or please give an explanation. If authors prefer the original authorship, a deep check on the authorship investigation will be given. The check will take time, we hope for your understanding sincerely.
Authors’ response: We have revised the authorship to keep only one first author.
Response to the Reviewer-1 comments
- I think it is an interesting work on COVID-19 vaccines and will bring clarity and arguments to those who read and use it.
Authors’ response: We express our gratitude to the reviewer’s affirmation of the study’s novelty and strength.
- The authors establish in the title the type of design they use, but later, in the "Methods" section, they do not mention it explicitly, something that they do in the abstract. From my point of view, it should be mentioned in "Methods" as well.
Authors’ Response: Thank you. We have revised the methods section to add the study design. Please see the revised methods section on page 3.
- In the "Methods" section, subsection 2.1.1. "study participants" (page 3) , I think the sentence could be improved if instead of saying "during the delta variant" it was said "since the delta variant emerged"
Authors’ response: Thank you. The sentence has been revised. Now it reads: “This meta-analysis scope comprises of vaccinated or unvaccinated populations that were infected with COVID-19 since the delta variant emerged.” Please see page 3 (2.1.1 sub-section).
- In subsection 2.1.4. "study screening and selection" (page 3), two acronyms are used (RAM and MAR) which, as I have been able to deduce, correspond to the initials of two of the authors of the manuscript. To avoid confusion, it should be made more explicit.
Authors’ Response: Thank you for picking these up. We have revised this sentence. Now it reads: “RAM carried out and recorded the above process and shared the record with other two authors (MAA and MAR) for verification. Discrepancies were discussed and resolved by consensus”. Please see page 4.
Response to the Reviewer-2 comments
- This is very interesting manuscript with meta-analysis on the effectiveness of COVID-19 delta variant on non-vaccinated and vaccinated population. The number size is good enough to conclude the results.
Authors’ response: Thank you.
- Is it possible to apply computational or statistical measurements to show the effectiveness of delta variant on both vaccinated and non- vaccinated population through global mapping?
Authors’ Response: We appreciate the reviewer’s advice. In this meta-analysis, eleven eligible studies were selected based on study objectives and inclusion criteria. These studies were conducted in seven countries (i.e., One study in the United Kingdom, four studies in USA, two studies in China, one study from each country: Chile, Portugal, Canada and Qatar). Please see Table 1. The authors’ used random effects meta-analysis models to obtain pooled two-doses COVID-19 vaccines effectiveness (VE) estimates against Delta variant (B.1.617.2) complemented with a sensitivity analysis to examine the effects of outliers. VE was calculated as 1 – risk ratio (RR). We did not extend these analytical explorations through global mapping due to lack of sufficient studies. The authors may take the opportunity as future research. Due to varying level of acceptance of the vaccine due to access issues, sociocultural, political, and economic, extrapolating to global mapping may produce supoirious findings as included studies do not represent low and middle incomoce countries.
- Also, an inclusion of a table based on which vaccine has been reported more effective against delta variant? Because this one of the great discussion now days about the vaccine effectiveness.
Authors’ Response: Thank you for the reviewer's suggestions. We selected the starting date of date of May 1st 2021 because it marks a period when B.1.617.2 was escalated variant of concern (VOC) having been earlier detected in April 20211,2. Eligible studies were selected if they were original articles; published between May1, 2021 and September 30, 2021 (focuses on delta variant); and focused on effectiveness of COVID-19 vaccines against delta variant (B.1.617.2) clinical perspective (please see page 3). Unfortunately, we did not find a sufficient number of studies to perform vaccine-specific explorations during this period. We highlighted this concern in the limitation section. It reads: “Studies on single-dose vaccination and its efficacy were excluded in this meta-analysis due to lack of reported data, lower effectiveness with the delta variant, insufficient number of studies, and we did not employ any analysis considering the sub-groups of different types of vaccines due to insufficient number of studies and reported combined results with more than one vaccine.” Please see page 9.
- It would be nice to add a short study on the effects of delta variant in post vaccinated population (prefer to present data through mapping).
Authors’ Response: Thank you for the reviewer's suggestions. The primary purpose of this meta-analysis is to estimate the pooled effectiveness of the COVID-19 vaccines against delta variant in terms of risk ratio (RR) among fully vaccinated, compared to unvaccinated populations. Eligible studies were selected based on study objectives and inclusion criteria, published between May1, 2021 and September 30, 2021 (focuses on delta variant); and focused on effectiveness of COVID-19 vaccines against delta variant (B.1.617.2) clinical perspective. Nontheless, we shall take the opportunity to investigate the effects of delta variant in post-vaccinated populations as a future research project.
- Overall, manuscript is well-written and data also presented quite well. Possibly after these changes manuscript would be highly interesting for the readers.
Authors’ response: We express our gratitude to the reviewer’s affirmation of the study’s novelty and strength.
References
- Public Health England. SARS-CoV-2 variants of concern and variants under investigation in England: Technical briefing 18 [Internet]. Public Health England, Wellington House, 133-155 Waterloo Road, London SE1 8UG: 2021. Available from: https://assets.publishing.service.gov.uk/government/uploads/system/uploads/attachment_data/file/990339/Variants_of_Concern_VOC_Technical_Briefing_13_England.pdf
- Noori M, Nejadghaderi SA, Arshi S, et al. Potency of BNT162b2 and mRNA-1273 vaccine-induced neutralizing antibodies against severe acute respiratory syndrome-CoV-2 variants of concern: A systematic review of in vitro studies. Rev Med Virol 2021;e2277:1–23.